# 880 nm NIR-Triggered Organic Small Molecular-Based Nanoparticles for Photothermal Therapy of Tumor

**DOI:** 10.3390/nano11030773

**Published:** 2021-03-18

**Authors:** Yunying Zhao, Zheng He, Qiang Zhang, Jing Wang, Wenying Jia, Long Jin, Linlin Zhao, Yan Lu

**Affiliations:** 1School of Materials Science & Engineering, Tianjin Key Laboratory for Photoelectric Materials and Devices, Key Laboratory of Display Materials & Photoelectric Devices, Ministry of Education, Tianjin University of Technology, Tianjin 300384, China; 17853483674@163.com (Y.Z.); 13752723282@163.com (Z.H.); zhangqiang@email.tjut.edu.cn (Q.Z.); wangjing@iccas.ac.cn (J.W.); jia15249238411@163.com (W.J.); KimYong0205@163.com (L.J.); luyan@tjut.edu.cn (Y.L.); 2State Key Laboratory of Molecular Engineering of Polymers, Fudan University, Shanghai 200433, China

**Keywords:** photothermal therapy, NIR-triggered, photothermal agent, deep tissue, nanoparticles

## Abstract

Photothermal therapy (PTT) has received constant attention as an efficient cancer therapy method due to locally selective treatment, which is not affected by the tumor microenvironment. In this study, a novel 880 nm near-infrared (NIR) laser-triggered photothermal agent (PTA), 3TT-IC-4Cl, was used for PTT of a tumor in deep tissue. Folic acid (FA) conjugated amphiphilic block copolymer (folic acid-polyethylene glycol-poly (β-benzyl-L-aspartate)_10_, FA-PEG-PBLA_10_) was employed to encapsulate 3TT-IC-4Cl by nano-precipitation to form stable nanoparticles (TNPs), and TNPs exhibit excellent photothermal stability and photothermal conversion efficiency. Furthermore, the in vitro results showed TNPs display excellent biocompatibility and significant phototoxicity. These results suggest that 880 nm triggered TNPs have great potential as effective PTAs for photothermal therapy of tumors in deep tissue.

## 1. Introduction

Phototherapy has attracted extensive attention in recent years as a powerful cancer treatment method due to characteristics such as convenience, noninvasiveness, locally selective treatment, negligible drug resistance and minimized adverse side effects [1]. Photodynamic therapy (PDT) and photothermal therapy (PTT) are two typical phototherapy approaches, PTT is based on the photothermal agents (PTA), which are preferentially taken up and retained by diseased tissue; then after excitation by appropriate wavelength laser, the PTA convert light to heat to induce cancer cell apoptosis or necrosis. Compared to PDT, PTT is not affected by the tumor microenvironment, such as the local oxygen level, so PTT has received increasing attention and developed rapidly in recent years.

PTAs are one of the most important factors determining the efficiency of PTT, and many kinds of PTA have been developed in recent years. Current PTAs can be classified as inorganic and organic materials, and compared to inorganic PTAs, the organic PTAs with easy chemical structure tuning, good biocompatibility, low-toxicity and an easy metabolism in the biological system are more desirable for clinical photo-theranostics [2,3,4,5,6,7], such as cyanine dyes [8,9,10,11,12,13], diketopyrrolopyrrole derivatives [14,15], croconaine-based agents [16,17], porphyrin-based agents [18,19,20,21], conjugated polymers [22,23,24,25,26,27,28,29], squaraine derivatives [30,31], boron dipyrromethane (BODIPY) dyes [32] and so on. In organic PTAs, the polymeric PTA was limited due to its complicated fabrication processes, indistinct biodegradation and potential biosafety [21]. Therefore, the small organic molecules have received increasing attention as potential alternatives to nanomaterials in the area of PTT recently.

In addition, another main challenge for phototherapy is to efficiently treat cancers at a deep tissue level. Near-infrared (NIR) light is referred to as the “optical window” of the biological tissues due to the minimal light absorption and scattering. Compared with the UV or visible light, NIR shows larger penetration distance in tissue, lower photodamage effect and higher signal-to-noise ratio [33,34]. The organic molecules with extended π-conjugation usually show strong NIR absorbance, which is beneficial for deep tumor tissue diagnosis and phototherapy [35,36,37]. The well-designed, conjugated small molecules of organic PTA, especially the recently reported acceptor-donor-acceptor (A-D-A) structure PTA, would open a new gate for efficient PTT of tumor in deep tissues [38,39,40].

However, a problem limiting the use of conjugated small molecules of organic PTA is their low water solubility; the hydrophobic PTAs are difficult to use to prepare pharmaceutical formulations and cannot be directly injected intravenously. To overcome these problems, various strategies have been employed to prepare water-soluble and stable formulations of hydrophobic organic PTA, such as conjugate to water-soluble polymers [11], loaded into mesoporous materials [19] or carbon materials [41,42,43], encapsulate in colloidal carriers such as liposomes [18] and polymer nanoparticles [9,10,14,15,20,21,22,23,24,32,44,45].

In this study, an A-D-A structure non-fullerene molecule, 3TT-IC-4Cl, which includes three fused thieno[3,2-b]thiophene as the central core and difluoro-substituted indanone as the end group was selected as PTA for PTT. Similarly to other A-D-A structure non-fullerene molecules, 3TT-IC-4Cl exhibits both broad absorption and effectively suppressed fluorescence [39], and especially, 3TT-IC-4Cl exhibits strong and broad absorption in the 800–900 nm region after forming nanoparticles, and it is indicated that the 3TT-IC-4Cl has the potential as PTA for NIR-triggered PTT of cancer in deep tissue. In order to effectively utilize 3TT-IC-4Cl for PTT, herein, our previous reported folic acid (FA) conjugated amphiphilic block copolymer (folic acid-polyethylene glycol-poly (β-benzyl-L-aspartate)_10_, FA-PEG-PBLA_10_) was employed to encapsulate 3TT-IC-4Cl by nano-precipitation and dialysis process to form stable nanoparticles (TNPs) and improve 3TT-IC-4Cl solubility in aqueous solution. In the TNPs system, the 3TT-IC-4Cl and PBLA segment of the copolymer was an inner core for 3TT-IC-4Cl storage, 3TT-IC-4Cl was the heat source and the PEG segment was the outer shell to improve solubility, stability and biocompatibility of this system, and the active targeting ligand FA was introduced to the surface of nanoparticles to enhance the selectivity of nanoparticles.

Recently, the NIR-triggered organic small molecular based PTT systems have been developed [9,10,14,15,19,24,32,46]; however, few systems of A-D-A type small molecular organic PTA-based and 880 nm-triggered PTT have been reported.

## 2. Materials and Methods

### 2.1. Materials

Folic Acid (FA), PEG-bis(amine) (Mn: 3.4 kDa), β-benzyl-L-aspartate (BLA), Triethylamine (TEA), Thiazolyl Blue Tetrazolium Bromide (MTT), Phosphate Buffered Saline (PBS), and Sodium Bicarbonate were purchased from Sigma Chemical Co. (St. Louis, MO, USA). Triphosgene was purchased from Aldrich Chemical Co. (Milwaukee, WI, USA). N-hydroxysuccinimide (NHS) and *N*,*N*’-dicyclohexylcarbodiimide were purchased from Fluka (Buchs, Switzerland). Then, 3TT-IC-4Cl was provided by Zhongsheng Huateng Technology Co., Ltd. (Beijing, China) according to a previously reported method [47]. Indocyanine Green (ICG) was purchased from Adamas (Shanghai, China). CHCl_3_ was purchased from Sinopharm Chemical Reagent Co., Ltd. (Shanghai, China). Dimethyl sulfoxide (DMSO) was purchased from Fuchen Chemical Reagent Co., Ltd. (Tianjin, China). Chloroform-d was purchased from Tenglong Weibo Technology Co., Ltd. (Qingdao, China). DMSO-d_6_ was purchased from Ningbo Cuiying Chemical Technology Co., Ltd. (Ningbo, China). Dulbecco’s modified Eagle’s medium (DMEM), Fetal Bovine Serum (FBS), Penicillin and Streptomycin were purchased from Gibco BRL (Invitrogen Corp., Carlsbad, CA, USA). All other chemicals were of an analytical grade and used as received without further purification.

### 2.2. Characterization

The chemical structure was determined by 400 MHz ^1^H NMR (AVANCE III HD 400 MHz, Bruker, Fällanden, Switzerland) using CHCl_3_-d and DMSO-d_6_ as the solvent. The photophysical properties of samples in aqueous solution were confirmed by UV-visible spectrophotometry (UV-2550, Shimadzu, Tokyo, Japan) and fluorescence spectrophotometer (F-4600, Hitachi, Tokyo, Japan). The morphologies, sizes and size distributions of nanoparticles were determined by transmission electron microscopes (TEM) (TECNAI G2 Spirit TWIN, FEI, Hillsboro, FL, USA) and dynamic light scattering (DLS) (Zetasizer Nano ZS90, Malvern Instruments Co, Malvern, UK) at 25 °C using a He-Ne laser (633 nm) as a light source. The temperature was monitored by IR thermal camera (TiS65, Fluke, Everett, WA, USA). The NIR laser (880 nm) used in this study was purchased from Beijing Laserwave Optoelectronics Technology Co., Ltd. (LWIRL880-20W-F, Laserwave, Beijing, China).

### 2.3. Preparation of TNPs

In order to prepare TNPs, first, the amphiphilic block copolymer FA-PEG-PBLA_10_ used for 3TT-IC-4Cl encapsulation was synthesized by ring-opening polymerization as our previous reported [48]. The chemical structure of FA-PEG-PBLA_10_ was confirmed by ^1^H NMR (400 MHz, DMSO). Then, the TNPs were prepared by the nanoprecipitation method. Briefly, 5 mg 3TT-IC-4Cl was dissolved in 1 mL THF; then, the 3TT-IC-4Cl solution was added into to 50 mL FA-PEG-PBLA_10_ solution (0.5 mg/mL in DMSO) dropwise, and then the mixture was transfered to dialysis tubs (Cut-off 3.5 K Mw) to remove THF and DMSO, followed by freeze drying, after which the TNPs were obtained.

### 2.4. Photothermal Effect

To confirm the PTT application potential, the photothermal property of TNPs was investigated, and a series of concentrations of TNPs (0, 30, 90, 180 and 250 μg/mL) in water were irradiated by 880 nm laser (0.7 W/cm^2^, where, the power densities (W/cm^2^) = laser beam power/laser beam area) for 720 s, the temperature of TNPs solution was recorded by an IR thermal camera every 30 s. In addition, the constant concentration (180 μg/mL) of TNPs were irradiated by an 880 nm laser for 720 s with various power densities (0.3, 0.5, 0.8 and 1.5 W/cm^2^) was investigated by the same method.

### 2.5. Stability of TNPs

In order to investigated the stability of TNPs, TNPs (180 μg/mL, 30 μg/mL free 3TT-IC-4Cl equiv.) and free ICG (30 μg/mL) were irradiated with an 880 nm laser (0.7 W/cm^2^) for 5 min; then the laser was turned off and the sample was cooled to the room temperature naturally, and the temperature of samples was recorded using the IR thermal camera every 30 s. Subsequently, the procedures were repeated four times.

### 2.6. In Vitro Phototoxicity and Biocompatibility of TNPs

HeLa cells (provided by Dingguo Biology Technology Co., Ltd., 1 × 10^4^ cells/well) were seeded onto 96-well plates in 200 μL DMEM and allowed to attach for 24 h. After cell attachment, the medium was replaced with 100 μL of fresh medium containing FA-PEG-PBLA_10_ (the polymer dispersed in aqueous medium) and TNPs with a series of concentration (0, 30, 60, 90, 120, 180 and 250 μg/mL), and then incubated for 4 h. The cells were washed with PBS and replace with fresh DMEM. The samples were irradiated with a laser (880 nm, 0.7 mW/cm^2^) for 5 min. Then, irradiated cells were incubated at 37 °C for 24 h and cell viability was evaluated by MTT assay. Data presented are averaged results of quadruplicate experiments. For biocompatibility, HeLa cells (1 × 10^4^ cells/well) were seeded onto 96-well plates in 200 μL DMEM and allowed to attach for 24 h. After cell attachment, the medium was replaced with 100 μL of fresh medium containing FA-PEG-PBLA_10_ and TNPs with a series of concentration (0, 30, 60, 90, 120, 180 and 250 μg/mL), and then they were incubated for 24 h. The cell viability was evaluated by an MTT assay. Data presented are averaged results of quadruplicate experiments.

## 3. Results and Discussion

### 3.1. Synthesis and Characterization of TNPs

A novel PTA with an 880 nm-triggered A-D-A structure non-fullerene molecule, 3TT-IC-4Cl, which included three fused thieno[3,2-b]thiophene as the central core and difluoro substituted indanone as the end group [47] was selected for PTT. In order to effectively utilize 3TT-IC-4Cl for tumor therapy. An amphiphilic block copolymer (FA-PEG-PBLA_10_) was synthesized as in our previous reported method [48] and used for 3TT-IC-4Cl encapsulation, 3TT-IC-4Cl was encapsulated in FA-PEG-PBLA_10_ by nano-precipitation and a dialysis process to form stable nanoparticles (TNPs), as shown in Figure 1, the PBLA segment of the copolymer was used as a reservoir for 3TT-IC-4Cl storage in the inner core, the PEG segment was used as the outer shell to improve solubility, stability and biocompatibility of TNPs, the active targeting ligand FA was introduced to the surface of nanoparticles to enhance selectivity of nanoparticles, the chemical structure was confirmed by ^1^H NMR, as shown in Figure 2A, and the characteristic peaks a and b are belong to FA-PEG-PBLA_10_, and the characteristic peaks c, d, e, f, g and h attribute to 3TT-IC-4Cl, respectively. It indicated that the 3TT-IC-4Cl was encapsulated in FA-PEG-PBLA_10_ successfully, the encapsulation rate (93.5%) was calculated by the relative intensity ratio of the methylene proton of PEG at 3.5 ppm and the proton of the alkane chain of in 3TT-IC-4Cl at about 1 ppm.

For nanomedicine used in cancer therapy, size, morphology and stability are the key properties that influence in vivo performance. These factors affect the bio-distribution and circulation time of the drug carriers. Stable and suitable-sized particles have reduced uptake by the reticuloendothelial systems (RES) and provide efficient passive tumor targeting ability via an enhanced permeation and retention (EPR) effect [49]. The incomplete tumor vasculature results in leaky vessels with gap sizes of 100 nm to 2 μm depending on the tumor type, and some studies have shown that particles with diameters of <200 nm are more effective [49,50]. The morphology of TNPs was evaluated by TEM, as shown in Figure 3. The TNPs were submicron in size and uniform and nearly spherical with no aggregation between nanoparticles observed due to the polymer modification, the average diameter was 150 nm. DLS measurements showed average hydrodynamic diameters of TNPs were about 200 nm (Figure 3, inset), a suitable size for passive targeting ability through the EPR effect. The size distribution of TNPs maintained a narrow and monodisperse unimodal pattern. Zeta potential of TNPs was measured as shown in Appendix A. It was shown that TNPs have negative surface charges, and zeta potential is about −13.2 mV. The zeta potential of TNPs showed that it would more stable against aggregation. Furthermore, the size of TNPs in DMEM remains almost same within 60 days (Appendix A).

### 3.2. Optical Properties of TNPs

The optical properties of TNPs were investigated by UV-vis absorption spectra and fluorescence spectra (Figure 4A,B), for free 3TT-IC-4Cl in CHCl_3_ solution, and it shows strong absorption at 772 nm and a maximal fluorescence at about 840 nm. However, after the formation of nanoparticles, the TNPs aqueous solution exhibits strong absorption at 874 nm, the significant red shift was due to the π-π stacking of 3TT-IC-4Cl during the nanoparticles formation and this result would be conducive to trigger TNPs by an 880 nm NIR light source for the phototherapy of the tumor in deep tissue. On the other hand, compared to free 3TT-IC-4Cl in CHCl_3_ solution, in the TNPs aqueous solution, nearly no fluorescence signal was observed due to the 3TT-IC-4Cl aggregation during the nanoparticle formation, which would significantly increase non-radiative heat generation and enhance PTT efficiency [20,51].

### 3.3. Photothermal Properties of TNPs In Vitro

To investigate the photothermal conversion property of the TNPs, the temperature of TNP aqueous solution with a series of concentrations (from 0 to 250 μg/mL) under the 880 nm laser irradiation (0.7 W/cm^2^) for 15 min was monitored (Figure 5A), and the related infrared (IR) thermal images of TNPs aqueous solution were showed in Figure 5C. As shown in the Figures, the temperature increased significantly as TNP concentration increased. It is noted that the TNPs at 90 μg/mL exhibit effective hyperthermia (>50 °C), which is sufficient to induce apoptosis or necrosis of cancer cells [52]. The relationship between temperature of TNPs aqueous solution (180 μg/mL) and different laser power (from 0.3 to 1.5 W/cm^2^) was future measured, as shown in Figure 5B, and the temperature of the TNPs aqueous solution depends on the laser power. The related infrared (IR) thermal images of TNP aqueous solution were showed in Figure 5D. On the other hand, we also investigated the photothermal conversion efficiency of TNPs through a cycle of heat-up and cooling using the previously reported method (Appendix A) [53]. The photothermal conversion efficiency of the TNPs was 31.5%, which is higher than other PTAs such as cyanine dyes (e.g., ≈26.6%) and gold nanorods (e.g., ≈21.0%) [24,54,55]. The strong absorption and high photothermal conversion efficiency of TNPs in the NIR region provided the potential of photothermal treatment of cancer.

### 3.4. Photothermal Stability of TNPs

The photothermal stability is an important parameter of photothermal drugs for PTT applications, and it would be crucial for clinical applications and therapeutic efficiency. The photothermal stability of TNPs was evaluated by monitoring its ability to maintain the temperature elevation. As shown in Figure 6A, the TNPs were irradiated at 0.7 W/cm^2^ for 5 min, then the laser was turned off, the following samples were cooled down to room temperature, the temperature was recorded by IR thermal camera throughout the process, this irradiation/cooling procedures were repeated five times, as Figure 6A shows, and TNPs displayed negligible change in their temperature elevation after five irradiation/cooling cycles. However, the temperature elevation of free ICG decreased significantly after one irradiation/cooling cycle. On the other hand, we also observed the changes in the color of the samples, as shown in Figure 6B, and after 5 min irradiation the color of free ICG solution changed observably, but the TNPs exhibit no change after 30 min irradiation. These results indicated the TNPs exhibit excellent photothermal stability.

### 3.5. In Vitro Cell Test

In order to investigate the feasibility of TNPs as nano photothermal agents for PTT, in vitro cytotoxicity of TNPs was investigated by MTT assay and the average cell viability was monitored. For a biocompatibility test, the dark toxicity of TNPs was investigated. As shown in Figure 7A, both FA-PEG-PBLA_10_ and TNPs exhibited no significant dark toxicity. As the concentration increased, the average cell viability was greater than 90% even when cells were treated with 250 μg/mL of TNPs. For the phototoxicity test, we investigated the concentration dependent (0, 30, 60, 90, 120, 180 and 250 μg/mL) cytotoxicity of TNPs with 880 nm laser irradiation. As shown in Figure 7B, after irradiation at 0.7 W/cm^2^ for 5 min, the cell viability gradually decreased as the TNPs concentration increased. Taken together, these results indicate that the TNPs could considerably enhance the efficiency of PTT for tumor in deep tissue, even at low concentrations.

## 4. Conclusions

In summary, an 880 nm NIR laser that triggered TNPs as PTA for photothermal therapy of a tumor in deep tissue was developed. In this work, a novel PTA, 3TT-IC-4Cl, was selected and used for PTT; it included three fused thieno[3,2-b]thiophene as the central core and difluoro-substituted indanone as the end group. After encapsulation by the FA-PEG-PBLA_10_ block copolymer and forming nanoparticles, the TNP aqueous solution exhibited strong absorption at 880 nm due to the π-π stacking. DLS and TEM measurements showed that the TNPs have a spherical shape and narrow size distribution with a mean diameter of 150 nm. TNPs exhibit excellent photothermal stability and high photothermal conversion efficiency after 880 nm laser irradiation. In the in vitro test, TNPs display excellent biocompatibility and significant phototoxicity. Therefore, the 880 nm-triggered TNPs have great potential as an effective PTA for the photothermal therapy of tumor in deep tissue.

## Figures and Tables

**Figure 1 nanomaterials-11-00773-f001:**
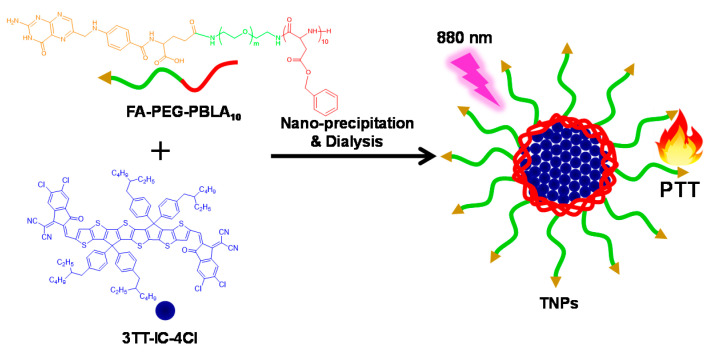
Schematic illustration demonstrating of stable nanoparticles (TNPs) formation and photothermal therapy (PTT) effect.

**Figure 2 nanomaterials-11-00773-f002:**
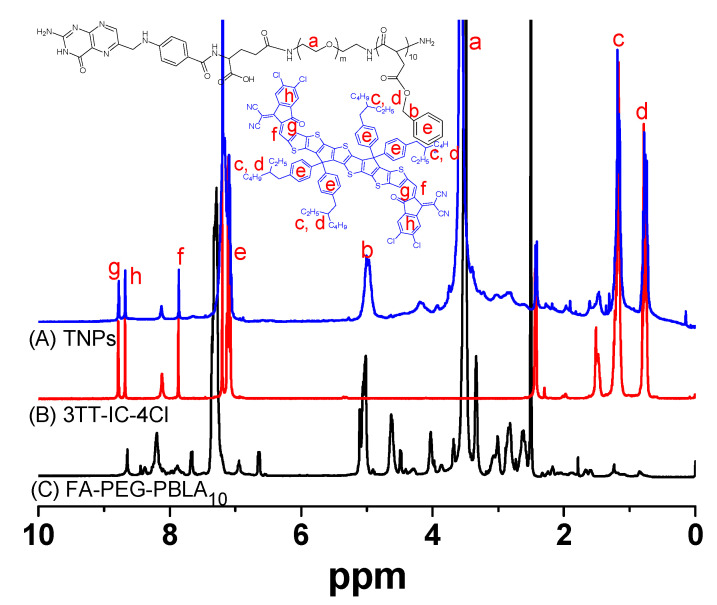
^1^H NMR spectra of (**A**) TNPs, (**B**) 3TT-IC-4Cl, and (**C**) FA-PEG-PBLA_10_.

**Figure 3 nanomaterials-11-00773-f003:**
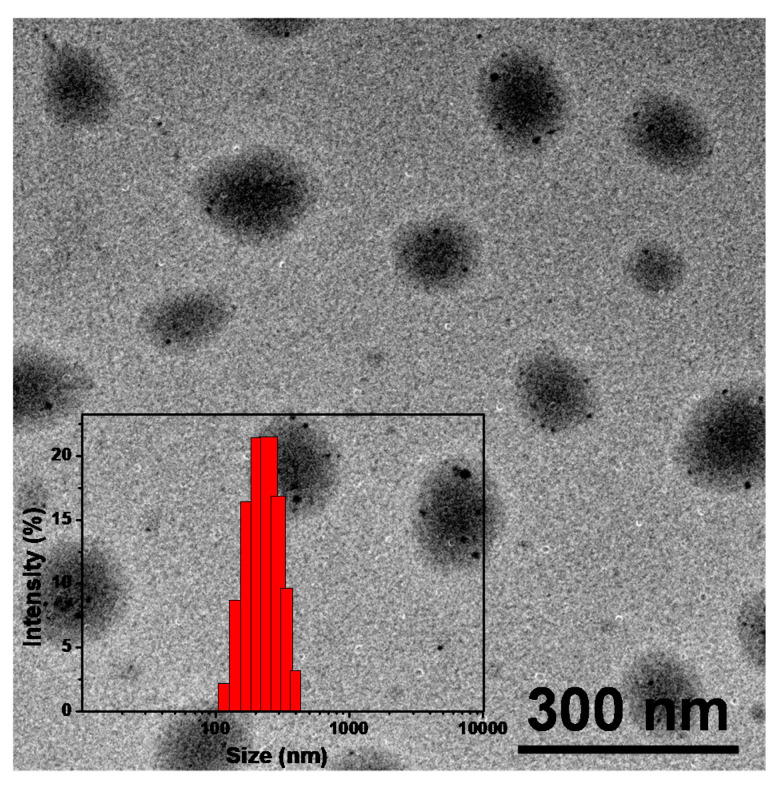
TEM image of TNPs and typical size distributions of TNPs (insert).

**Figure 4 nanomaterials-11-00773-f004:**
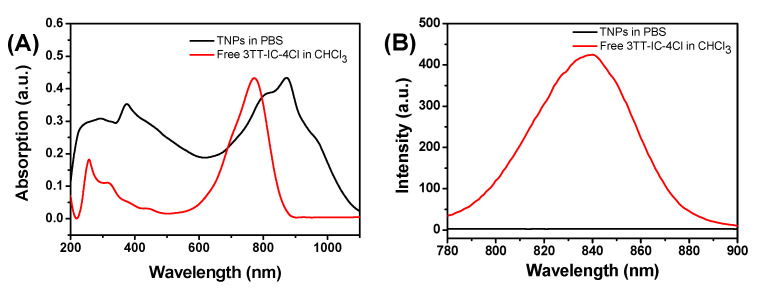
(**A**) UV–Vis absorption spectra of free 3TT-IC-4Cl (red) and TNPs (black), and (**B**) Fluorescence spectra of free 3TT-IC-4Cl (red) and TNPs (black).

**Figure 5 nanomaterials-11-00773-f005:**
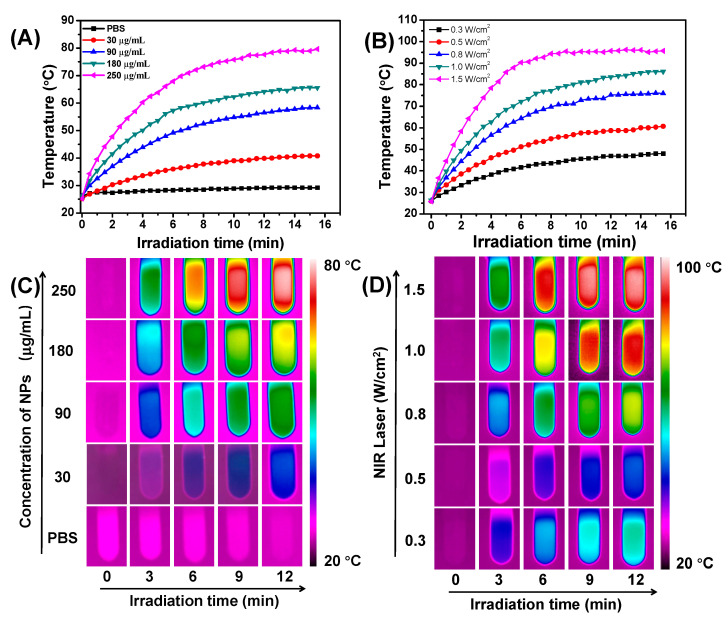
(**A**) Photothermal conversion behavior of TNPs at different concentrations (0–250 μg/mL) under 880 nm irradiation at 0.7 W/cm^2^, (**B**) Photothermal conversion behavior of TNPs at different laser power (0.3–1.5 W/cm^2^) under 880 nm irradiation at 0.7 W/cm^2^, and (**C**) IR thermal images of TNPs at different concentrations (0–250 μg/mL) under 880 nm irradiation at 0.7 W/cm^2^, and (**D**) IR thermal images of TNPs at different laser power (0.3–1.5 W/cm^2^) under 880 nm irradiation at 0.7 W/cm^2^.

**Figure 6 nanomaterials-11-00773-f006:**
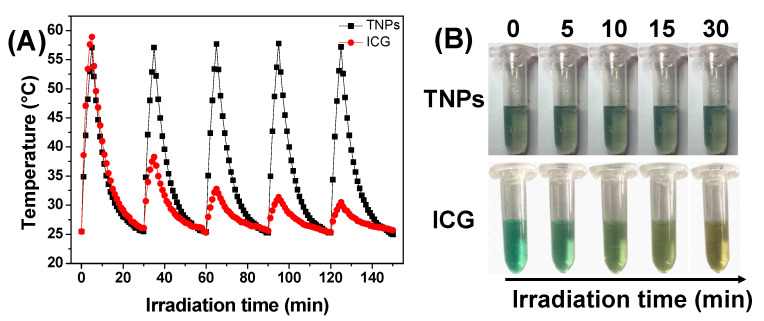
(**A**) Temperature elevation of TNPs, and free ICG under five irradiation/cooling cycles (under 880 nm irradiation at 0.7 W/cm^2^ for 5 min), (**B**) Photographs of the TNPs, and free ICG in PBS solutions after 880 nm light irradiation for different time.

**Figure 7 nanomaterials-11-00773-f007:**
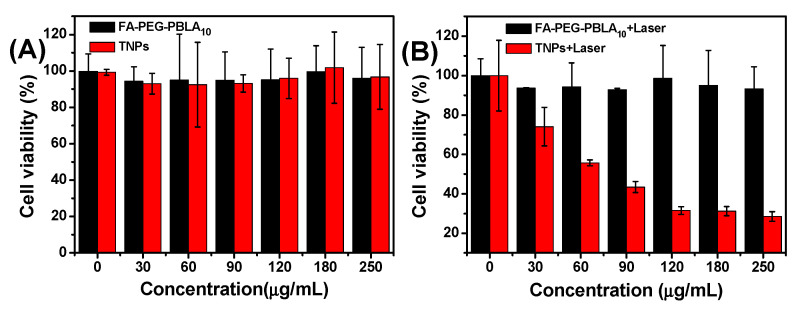
In vitro cytotoxicity test using FA-PEG-PBLA_10_ and TNPs against HeLa cells (**A**) dark toxicity depending on the nanoparticles concentration and (**B**) phototoxicity depending on nanoparticles concentration.

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
