# Peer review of "880 nm NIR-Triggered Organic Small Molecular-Based Nanoparticles for Photothermal Therapy of Tumor"

_nanomaterials, 2021, doi:10.3390/nano11030773_

Round 1

Reviewer 1 Report

This manuscript details the preparation and in vitro evaluation of folic acid-targeted poly(ethylene glycol-b-beta benzyl L-aspartate) nanoparticles for the delivery of the photothermal agent 3TT-IC-4Cl. The following changes will improve the quality of the manuscript:

  • Additional and more recent references should be provided in the introduction, specially for the use of cyanine dyes (which is written as “days”) and conjugated polymers as PTAs.
    • Dyes:
      • J. Biol. Macromol. 2020, 163, 156-166.
      • J. Pharm. 2020, 589, 119787
    • Conjugated Polymers:
      • ACS Appl. Polym. Mater. 2020, 2, 4258-4272 (Review)
      • ACS Appl. Polym. Mater. 2020, 2, 4222-4240 (Review)
      • ACS Appl. Polym. Mater. 2020, 2, 4180-4187
      • ACS Appl. Polym. Mater. 2020, 2, 5602-5620
      • J. Biol. Macromol. 2021, 166, 98-107.
  • Define abbreviations on first use (MTT, PBS, DMSO)
  • Please specify the laser beam width as well as how the laser fluence (W/cm2) was measured. This information should be included in the methods section.
  • What is the difference between FA-PEG-PBLA10 and TNPs described in sections 2.6 and 3.5? Is FA-PEG-PBLA the polymer in the form of nanoparticles but without the dye or is it just the polymer dispersed in aqueous medium? This needs to be explicitly clarified in the text. If it is dye-fee nanoparticles, then the characteristics of these particles will also need to be provided in the article (size by TEM and DLS, optical properties)
  • The chemical structure of the polymer in Figure 1 is inaccurate. Specifically, the poly(benzyl aspartate) needs to include an NH within the parenthesis to indicate that the repeat unit is a polypeptide derivative. You can do this by including the NH from the NH2 in the parenthesis and just leaving an H on the terminus.
  • The size of the nanoparticles is too large. The hydrodynamic size of the particles appears to be 200 nm in average, but the particle size range spans from 100 nm to ~ 450 nm according to the DLS plot.  It is well known that nanoparticles with diameter less than 200 nm (and ideally less than 100 nm) show enhanced behavior in vivo. Particles larger than 200 nm are very likely to be rapidly removed by the reticuloendothelial system and end up in the liver and spleen. This should be acknowledged in the manuscript and proper references cited in page 5. In addition, the authors should show that the nanoparticle size could (likely) be decreased by formulation optimization by showing the effect of polymer/dye concentration on the organic feed on the size of the particles. Finally, they should show the effect of freeze drying on particle size by showing the size before and after freeze/drying and resuspending as it is well known that PEGylated nanoparticles tend to aggregate through the freeze drying process.
  • The zeta potential of the dye-loaded and dye-free nanoparticles should be provided
  • Additional information that should be included for on the size of the nanoparticles in serum or at least in cell medium with serum, as well as the size stability over time in this medium
  • Since the use oof 3TT-IC-4Cl is key to the function of these particles, additional information must be provided for others to be able to reproduce the results of this work. Looking at the website of the manufacturing company (SunaTech), the dye 3TT-IC-4Cl does not appear in their catalog (at least not under this name). Please provide the chemical name, CAS number and chemical formula for this compound. Also, if possible, provide references for methods of the synthesis of this dye.
  • In the conclusion, please revise or remove the sentence that says “These stable nanoparticles are suitable for the EPR effect and accumulation in the tumor tissue” as this is not something that was demonstrated in this study.
  • English grammar and word choice needs to be improved throughout the manuscript.
  • Hyphenation throughout manuscript needs to be revised

Author Response

Point 1: Additional and more recent references should be provided in the introduction, specially for the use of cyanine dyes (which is written as “days”) and conjugated polymers as PTAs.Dyes: J. Biol. Macromol. 2020, 163, 156-166; J. Pharm. 2020, 589, 119787. Conjugated Polymers: ACS Appl. Polym. Mater. 2020, 2, 4258-4272 (Review); ACS Appl. Polym. Mater. 2020, 2, 4222-4240 (Review); ACS Appl. Polym. Mater. 2020, 2, 4180-4187; ACS Appl. Polym. Mater. 2020, 2, 5602-5620; J. Biol. Macromol. 2021, 166, 98-107.

Response 1: Thank you very much for your suggestion. The mentioned relevant literatures have been added in the revised manuscript (Ref: 12, 13, and 25 to 29).

Point 2: Define abbreviations on first use (MTT, PBS, DMSO)

Response 2: Thank you very much for your suggestion. The abbreviations on first use (MTT, PBS, DMSO) have been defined in the revised manuscript (line 83, 84 and 91).

Point 3: Please specify the laser beam width as well as how the laser fluence (W/cm2) was measured. This information should be included in the methods section.

Response 3: Thank you very much for your suggestion. The laser fluence (W/cm2) measure method have been specified in the revised manuscript (line 123).

Point 4: What is the difference between FA-PEG-PBLA10 and TNPs described in sections 2.6 and 3.5? Is FA-PEG-PBLA the polymer in the form of nanoparticles but without the dye or is it just the polymer dispersed in aqueous medium? This needs to be explicitly clarified in the text. If it is dye-fee nanoparticles, then the characteristics of these particles will also need to be provided in the article (size by TEM and DLS, optical properties).

Response 4: Thank you very much for your suggestion. In this study, the FA-PEG-PBLA10 is just the polymer dispersed in aqueous medium, it has been clarified in the revised manuscript (line 137).

Point 5: The chemical structure of the polymer in Figure 1 is inaccurate. Specifically, the poly(benzyl aspartate) needs to include an NH within the parenthesis to indicate that the repeat unit is a polypeptide derivative. You can do this by including the NH from the NH2 in the parenthesis and just leaving an H on the terminus.

Response 5: Thank you very much for your suggestion. The chemical structure of the polymer in Figure 1 have been modified.

Point 6: The size of the nanoparticles is too large. The hydrodynamic size of the particles appears to be 200 nm in average, but the particle size range spans from 100 nm to ~ 450 nm according to the DLS plot. It is well known that nanoparticles with diameter less than 200 nm (and ideally less than 100 nm) show enhanced behavior in vivo. Particles larger than 200 nm are very likely to be rapidly removed by the reticuloendothelial system and end up in the liver and spleen. This should be acknowledged in the manuscript and proper references cited in page 5. In addition, the authors should show that the nanoparticle size could (likely) be decreased by formulation optimization by showing the effect of polymer/dye concentration on the organic feed on the size of the particles. Finally, they should show the effect of freeze drying on particle size by showing the size before and after freeze/drying and resuspending as it is well known that PEGylated nanoparticles tend to aggregate through the freeze drying process.

Response 6: Thank you very much for your suggestion. Some views about EPR effect have been acknowledged and cited literature in the revised manuscript (page 6, line 175). As reviewer pointed out, if we can control particles with diameters <100 nm, they would exhibit more effective EPR effect. Normally, the nanoparticle size would be decreased by formulation optimization, such as control polymer/dye feed ratio to control the size of the particles, we have reported a similar work before (Journal of Biomaterials Applications, 2013, 28, 434-447). However, in this work, it is difficult to control the particles size by controlling polymer/dye feed ratio. As shown in table R1, the polymer/dye feed ratio can not affect the particles size significantly. The drug loading content, photothermal conversion efficiency and morphology were comprehensively considered, polymer/dye feed ratio as 5:1 was selected in this study. In the future work, we are planning to control the particles size by some other method, such as using different organic solvent or using different precipitation method.

Table R1. Nanoparticles size depending on polymer/dye feed ratio

Entry FA-PEG-PBLA10 / 3TT-IC-4Cl feed ratio (M / M) Hydrodynamic size (nm)
1 1:1 200
2 2:1 198
3 5:1 200
4 10:1 203

Point 7: The zeta potential of the dye-loaded and dye-free nanoparticles should be provided.

Response 7: Thank you very much for your suggestion. The zeta potential of the dye-loaded nanoparticles (TNPs) have been provided in Supporting Information file (figure S1). Because we did not select dye-free nanoparticles as the control sample, so we did not provide the zeta potential of dye-free nanoparticles.

Point 8: Additional information that should be included for on the size of the nanoparticles in serum or at least in cell medium with serum, as well as the size stability over time in this medium.

Response 8: Thank you very much for your suggestion. The changes of hydrodynamic diameters of TNPs in DMEM with time have been provided in Supporting Information file (figure S2).

Point 9: Since the use of 3TT-IC-4Cl is key to the function of these particles, additional information must be provided for others to be able to reproduce the results of this work. Looking at the website of the manufacturing company (SunaTech), the dye 3TT-IC-4Cl does not appear in their catalog (at least not under this name). Please provide the chemical name, CAS number and chemical formula for this compound. Also, if possible, provide references for methods of the synthesis of this dye.

Response 9: Thank you very much for your suggestion. I am sorry that we have confused the 3TT-IC-4Cl manufacturing company. The 3TT-IC-4Cl was provided by Zhongsheng Huateng Technology Co., Ltd (Beijing, China, an organic synthesis company) according previous reported method (Advanced Science 2018, 5, 1800307). This information has been provided in the revised manuscript (line 88).

Point 10: In the conclusion, please revise or remove the sentence that says “These stable nanoparticles are suitable for the EPR effect and accumulation in the tumor tissue” as this is not something that was demonstrated in this study.

Response 10: Thank you very much for your suggestion. The sentence “These stable nanoparticles are suitable for the EPR effect and accumulation in the tumor tissue” has been removed in the revised manuscript (line 273).

Point 11: English grammar and word choice needs to be improved throughout the manuscript.

Response 11: Thank you very much for your suggestion. The English grammar and word choice have been improved in the revised manuscript.

Point 12: Hyphenation throughout manuscript needs to be revised.

Response 12: Thank you very much for your suggestion. The hyphenation throughout manuscript have been revised in the manuscript.

Reviewer 2 Report

Ref Manuscript ID: nanomaterials-1124911

The article entitled “880 nm NIR-Triggered Organic Small Molecular Based Nanoparticles for Photothermal Therapy of Tumor” by Yunying Zhao et al. describes a novel 880 nm near-infrared (NIR) laser triggered photothermal agent (PTA), 3 TT-IC-4Cl encapsulated in FA-PEG-PBLA10. The results indicate that TNPs exhibit excellent photothermal stability and photothermal conversion efficiency combined with increased biocompatibility and significant phototoxicity which make this class of agents suitable for in depth photothermal therapy of various tumor tissues.  The data are well presented, the experiments are systematically performed and the conclusions support the presented results. However, due to the misspelling and many grammatical errors, this manuscript is hard to be readable. Considering these, an extensive editing of English language and style is mandatory before any acceptance in Nanomaterials journal.

Few examples of grammatical errors are listed below:

Line 25: “pho-tothermal” must be corrected with “photothermal”

Line 29: “Compare to PDT” must be corrected with “Compared to PDT”

Line 30: “develop rapidly” must be corrected with “developed rapidly”

Line 33: “compare to PDT” must be corrected with “compared to PDT”

Line 35: “cyanine days” must be corrected with “cyanine dyes”

Line 42: “is efficient treat cancers” must be corrected with “is to efficiently treat cancers”

Line 43: “lev-el” must be corrected with “level”

Line 47: “benefi-cial” must be corrected with “beneficial”

Line 48: “mole-cules” must be corrected with “molecules”

And many others…..

Author Response

Point 1: The article entitled “880 nm NIR-Triggered Organic Small Molecular Based Nanoparticles for Photothermal Therapy of Tumor” by Yunying Zhao et al. describes a novel 880 nm near-infrared (NIR) laser triggered photothermal agent (PTA), 3TT-IC-4Cl encapsulated in FA-PEG-PBLA10. The results indicate that TNPs exhibit excellent photothermal stability and photothermal conversion efficiency combined with increased biocompatibility and significant phototoxicity which make this class of agents suitable for in depth photothermal therapy of various tumor tissues. The data are well presented, the experiments are systematically performed and the conclusions support the presented results. However, due to the misspelling and many grammatical errors, this manuscript is hard to be readable. Considering these, an extensive editing of English language and style is mandatory before any acceptance in Nanomaterials journal.

Few examples of grammatical errors are listed below:

Line 25: “pho-tothermal” must be corrected with “photothermal”

Line 29: “Compare to PDT” must be corrected with “Compared to PDT”

Line 30: “develop rapidly” must be corrected with “developed rapidly”

Line 33: “compare to PDT” must be corrected with “compared to PDT”

Line 35: “cyanine days” must be corrected with “cyanine dyes”

Line 42: “is efficient treat cancers” must be corrected with “is to efficiently treat cancers”

Line 43: “lev-el” must be corrected with “level”

Line 47: “benefi-cial” must be corrected with “beneficial”

Line 48: “mole-cules” must be corrected with “molecules”

And many others…..

Response 1: Thank you very much for your suggestion. The misspelling and grammatical errors have been improved in the revised manuscript.
